# Vitamin D Supplementation Regulates Postoperative Serum Levels of PD-L1 in Patients with Digestive Tract Cancer and Improves Survivals in the Highest Quintile of PD-L1: A Post Hoc Analysis of the AMATERASU Randomized Controlled Trial

**DOI:** 10.3390/nu13061987

**Published:** 2021-06-09

**Authors:** Makoto Morita, Mai Okuyama, Taisuke Akutsu, Hironori Ohdaira, Yutaka Suzuki, Mitsuyoshi Urashima

**Affiliations:** 1Division of Molecular Epidemiology, The Jikei University School of Medicine, 3-25-8 Nishi-Shimbashi, Minato-Ku, Tokyo 105-8461, Japan; external.mm@gmail.com (M.M.); maiokuyama0511@gmail.com (M.O.); taisuke0107.jusom@gmail.com (T.A.); 2Pfizer Japan Inc., Shinjuku Culture Quint Bldg, 3-22-7 Yoyogi, Shibuya-ku, Tokyo 151-8589, Japan; 3Department of Surgery, International University of Health and Welfare Hospital, 537-3 Iguchi, Nasushiobara, Tochigi 329-2763, Japan; ohdaira@iuhw.ac.jp (H.O.); yutaka@iuhw.ac.jp (Y.S.)

**Keywords:** vitamin D, supplement, PD-L1, cancer, serum, soluble, survival, randomized, placebo

## Abstract

Because vitamin D responsive elements have been found to be located in the PD-L1 gene, vitamin D supplementation was hypothesized to regulate serum PD-L1 levels and thus alter survival time of cancer patients. A post hoc analysis of the AMATERASU randomized, double-blind, placebo-controlled trial of postoperative vitamin D3 supplementation (2000 IU/day) in 417 patients with stage I to stage III digestive tract cancer from the esophagus to the rectum was conducted. Postoperative serum PD-L1 levels were measured by ELISA and divided into quintiles (Q1–Q5). Serum samples were available for 396 (95.0%) of the original trial. Vitamin D supplementation significantly (*p* = 0.0008) up-regulated serum PD-L1 levels in the lowest quintile (Q1), whereas it significantly (*p* = 0.0001) down-regulated them in the highest quintile (Q5), and it did not either up- or down-regulate them in the middle quintiles (Q2–Q4). Significant effects of vitamin D supplementation, compared with placebo on death (HR, 0.34; 95% CI, 0.12–0.92) and relapse/death (HR, 0.37; 95% CI, 0.15–0.89) were observed in the highest quintile (Q5) of serum PD-L1, whereas significant effects were not observed in other quintiles (P_interaction_ = 0.02 for death, P_interaction_ = 0.04 for relapse/death). Vitamin D supplementation significantly reduced the risk of relapse/death to approximately one-third in the highest quintile of serum PD-L1.

## 1. Introduction

Programmed death-ligand 1 (PD-L1) is expressed on a part of cancer cells to suppress anti-cancer immunity by interacting with the programmed death-1 (PD-1) receptor expressed on immune cells [1]. Indeed, blocking this interaction by administering monoclonal antibodies targeting either the PD-1 or the PD-L1 molecule improves the prognosis of patients with cancer, at least in part [2]. Moreover, PD-L1 is constitutively expressed at low levels on non-cancer cells, e.g., antigen-presenting cells, vascular endothelial cells, and pancreatic islet cells, which may induce immune tolerance by maintaining the quiescence of autoreactive immune cells [1].

Membrane-bound forms of PD-L1 are also expressed on the surface of exosomes, whereas soluble forms of PD-L1 are generated as splice variants or by proteolytic cleavage of membrane-bound forms, and both are secreted into the extracellular space and blood stream [3,4,5]. Thus, total levels of serum PD-L1 measured by enzyme-linked immunosorbent assay (ELISA) may reflect the sum of both exosomal and soluble forms of PD-L1. Serum PD-L1 was considered to be functional and shown in vitro to induce apoptosis of CD4+ and CD8+ T cells derived from either a patient with cancer or a healthy person [6,7]. In addition, we and our colleagues reported that serum PD-L1 levels were increased up to seven-fold in pregnant women compared with age-matched non-pregnant women, and further demonstrated in vitro that the increased serum PD-L1 of pregnant women suppressed both autogenic and allogeneic immune reactions, as well as cytokine production of immune cells [8]. In fact, a recent meta-analysis including a total of 21 studies demonstrated that elevated serum PD-L1 levels were associated with worse survival of patients with cancer [9]. In particular, higher postoperative, but not preoperative, plasma total PD-L1, in addition to exosomal PD-L1, was shown to be associated with poor survival in patients with gastric cancer [10]. Thus, not only relying on immune checkpoint inhibitors, but also reducing serum PD-L1 levels after operation, is another distinct strategy to improve the prognosis of patients with cancer. However, few strategies are suitable for clinical use at the moment except for therapeutic plasma exchange [11].

Vitamin D is a precursor of 1, 25(OH)D, which is a potent steroid hormone, and has been reported to have both positive and negative transcriptional regulations of gene expressions relating to innate immune responses through the vitamin D receptor in the target cell [12]. Of interest, vitamin D-responsive elements have been found to be located in an intronic region of the PD-L1 gene [13]. However, there are few reports of the interaction between serum PD-L1 and vitamin D. Vitamin D supplementation was hypothesized to regulate the serum levels of PD-L1 and thus change survival time of patients with cancer. We and our colleagues previously conducted the AMATERASU randomized, double-blind, placebo-controlled trial of postoperative vitamin D3 supplementation (2000 IU/day) in 417 patients with stage I to stage III digestive tract cancer from the oesophagus to the rectum who underwent curative surgery [14]. By conducting a post hoc analysis of the AMATERASU trial, the aim of this study was thus to examine the effects of vitamin D supplementation on the serum PD-L1 levels 1 year after starting supplements and on survival in each quintile of serum PD-L1 levels in patients with digestive tract cancer.

## 2. Materials and Methods

### 2.1. Trial Design

This study was a post hoc analysis of the AMATERASU trial (UMIN000001977) conducted in Japan, the details of which have been previously reported [14]. Briefly, 417 patients with digestive tract cancers from the oesophagus to the rectum participated in a randomized, double-blind, placebo-controlled trial to compare the effects of vitamin D3 supplements (2000 IU/day) and placebo on relapse and/or death at an allocation ratio of 3:2 at the International University of Health and Welfare Hospital (Otawara, Tochigi, Japan) between January 2010 and February 2018. The trial protocol was approved by the ethics committee of the International University of Health and Welfare Hospital (Otawara, Tochigi, Japan) (ethics approval code: 13-B-263), as well as the Jikei University School of Medicine (Nishi-shimbashi, Tokyo, Japan) (ethics approval code: 21-216 (6094)). Written, informed consent was obtained from each participating patient before surgery.

### 2.2. Participants

Details of the inclusion and exclusion criteria were described in the original report [14]. Briefly, the trial included patients not taking vitamin D supplements with stage I to stage III digestive tract cancers (esophageal, gastric, small intestinal, and colorectal) who underwent curative surgery with complete tumor resection. The outcome of relapse or death was confirmed by regular outpatient follow-up. The elapsed time to relapse or death was calculated from the time of randomization (i.e., time from starting the study supplements).

### 2.3. Measurement of Serum PD-L1 Levels

Serum samples for PD-L1 measurements were collected after the surgery (23 days, interquartile range (IQR): 13–43.5 days) and just before the start of vitamin D/placebo supplementation. The serum PD-L1 level was also measured 1 year after starting vitamin D/placebo supplements. The serum samples were stored at −80 °C prior to use. Serum PD-L1 levels were measured by a member of the research team, who was blind to randomized groups and clinical information including outcomes, which were fixed prior to statistical analyses, using ELISA kits from Abcam (#ab214565) (Cambridge, MA, USA), according to the manufacturer’s protocols. The lower detection limit for serum PD-L1 of the ELISA kit was 3.9 pg/mL, and the upper detection limit was 1300 pg/mL.

### 2.4. Evaluation of Other Covariates

The details of the analysis of histopathological subtypes [15], analysis of p53 protein, vitamin D receptor (VDR), Ki-67 by immunohistochemistry [16], and serum levels of bioavailable 25-hydroxyvitamin D (25(OH)D) [17] have been described in previous reports. Histopathological subtypes were not mutually exclusive because there could be multiple subtypes; p53-positive was defined as a positive nuclear percentage in the tumor epithelium greater than 10%. VDR was defined as a score using a semiquantitative scoring system, and Ki-67 was defined as the positive nuclear staining percentage in tumor epithelium. Bioavailable 25(OH)D was calculated using serum concentrations of 25(OH)D, vitamin D binding protein, albumin, and single-nucleotide polymorphisms of vitamin D binding protein.

### 2.5. Statistical Analysis

All patients who underwent randomization and for whom residual serum samples were available were included in this analysis. Spearman’s rank correlation coefficient (RHO) was used to quantify the strengths of associations between two continuous variables: RHO ≥ 0.4, strong; 0.4 > RHO ≥ 0.2, moderate; and RHO < 0.2, weak. Non-parametric continuous variables and dichotomous variables were compared between groups by the Mann–Whitney test and the chi-squared test, respectively. Changes in serum PD-L1 levels from baseline to 1 year later in either the vitamin D or placebo group were analyzed using the Wilcoxon signed-rank test.

Relapse and death-related outcomes were assessed according to the randomization group by whether or not supplements were taken. The effects of vitamin D and placebo on the risks of outcomes, i.e., total death and relapse/death, were estimated using Nelson–Aalen cumulative hazard curves. A Cox proportional hazards model was used to determine hazard ratios (HRs) and 95% confidence intervals (CIs) for the outcomes. To evaluate the effects of vitamin D supplementation on relapse, cumulative incidence functions were applied by considering patient deaths due to causes other than cancer relapse as a competing risk; competing risk regression was performed using subdistribution hazard ratios (SHRs) and 95% CIs [18]. When the 95% CI did not include 1, the HR and SHR were considered significant. To clarify whether vitamin D supplementation differed significantly among quintiles of serum PD-L1 levels (Q1–Q5), the *p* for interaction was analyzed on the basis of a Cox regression model including three variables (vitamin D group, the highest quintile of serum PD-L1 (Q5), and both the vitamin D group and the highest quintile (Q5) of serum PD-L1) by two-way interaction tests comparing the subgroup of the highest quintile of serum PD-L1 and the others. Values of *p* for interaction with two-sided *p* < 0.05 were considered significant. All data were analyzed using Stata 14.0 (StataCorp LP; College Station, TX, USA).

## 3. Results

### 3.1. Study Population

Of the 417 patients with digestive tract cancers who were randomly assigned to receive vitamin D supplements (*n* = 251, 60%) or placebo (*n* = 166, 40%), ELISA results for serum PD-L1 were available for 396 (95.0%) of the original AMATERASU trial participants (244 (97.2%) of the vitamin D group and 152 (91.6%) of the placebo group) because they were not sampled from patients or used up for other studies (Figure 1). However, 1 year after starting supplements, the number of available serum PD-L1 samples was further reduced to 319 (80.6%) (198 (81.1%) of the vitamin D group and 121 (79.6%) of the placebo group), due to death, transfer to other hospitals, sampled not being taken from patients, or samples being used up for other studies. The median follow-up of these 396 patients was 3.5 years (interquartile range (IQR): 2.4–5.4 years).

### 3.2. Patients’ Characteristics Stratified by Vitamin D Group and Placebo Group

Patients’ characteristics by vitamin D group and placebo group are shown in Table 1. The 3:2 ratio of assignment to the vitamin D and placebo groups was generally maintained for all variables. Of the 396 participants, 33% were women. The median age (IQR) was 66 (60–74) years, and the median body mass index was 21.9 (19.8–23.8) kg/m^2^. Percentages of cancer sites were as follows: esophageal, 9%; gastric, 42%; small intestinal, 1%; and colorectal, 48%. Disease stages were I, II, and III in 44%, 26%, and 30% of patients, respectively.

### 3.3. Serum PD-L1 Levels before and after Starting Supplements

Serum PD-L1 levels were assessed in 396 patients (Figure 2A). The median (IQR) level was 55.5 (44.2–70.2) pg/mL, with the distribution skewed to the right. Strong associations between serum PD-L1 levels before and after starting supplements were observed in the total sample (Figure 2B), in the vitamin D group (Figure 2C), and in the placebo group (Figure 2D). 

### 3.4. Patients’ Characteristics Stratified by Quintiles of Serum PD-L1 Levels

Patients’ characteristics in subgroups stratified by quintiles of serum PD-L1 levels are shown in Table 2. There were no differences in serum 25(OH)D levels or bioavailable 25(OH)D levels before vitamin D intervention among subgroups. Moreover, distributions of sex, body mass index, history of other cancers, comorbid conditions (except that a previous history of coronary artery disease was more frequent in Q5 than in other quintiles), site of cancers, stage, pathology, p53 expression, VDR expression, and adjuvant chemotherapy were also not different. There was only a difference for age, which was significantly higher in higher quintiles.

### 3.5. Effects of Vitamin D Supplementation on Serum PD-L1 Levels

The effects of vitamin D supplementation, as well as placebo, on serum PD-L1 levels were compared between pre (=after the surgery and just before starting supplement) and post (=1 year after starting supplement) supplementation in each quintile of the serum PD-L1 level (Figure 3). In the lowest quintile (Q1), vitamin D supplementation significantly up-regulated serum PD-L1 levels (*p* = 0.0008), with no significant change in the placebo group. On the other hand, in the highest quintile (Q5), vitamin D supplementation significantly down-regulated serum PD-L1 levels (*p* = 0.0001) despite no significant changes in the placebo group. On the other hand, vitamin D supplementation did not either up- or down-regulate serum PD-L1 levels in the middle quintiles (Q2, Q3, and Q4) and in the total sample (all quintiles).

### 3.6. Effect of the Interaction between Vitamin D Supplementation and Serum PD-L1 Quintiles on Hazard Risk of Death

First, the effects of vitamin D supplementation on HRs of death were compared among quintiles of serum PD-L1 levels (Figure 4). A significant effect of vitamin D, compared with placebo, was observed in the highest quintile (Q5) of serum PD-L1 (HR 0.34; 95% CI 0.12–0.92). On the other hand, significant effects of vitamin D on HRs of death were not observed in other quintiles, i.e., Q1 to Q4 and all quintiles excluding Q5 (Q1–Q4). There was a significant two-way interaction between the subgroup of Q5 and vitamin D supplementation (*p* for interaction = 0.04), even on multivariate adjustment with (1) age, (2) sex, (3) body mass index, (4) cancer sites, i.e., esophageal, gastric, and small intestinal plus colorectal cancers, (5) stage, (6) adjuvant chemotherapy, and (7) p53 positivity (*p* for interaction = 0.02).

### 3.7. Effect of the Interaction between Vitamin D Supplementation and Serum PD-L1 Quintiles on Hazard Risk of Relapse or Death

Next, the effects of vitamin D supplementation on HRs of relapse or death were compared among quintiles of serum PD-L1 levels (Figure 5). Similarly, a significant effect of vitamin D, compared with placebo, was observed in the highest quintile (Q5) of serum PD-L1 (HR 0.37; 95% CI 0.15–0.89). On the other hand, significant effects of vitamin D were not observed in other quintiles, i.e., Q1 to Q4 and all quintiles excluding Q5 (Q1–Q4). There was no significant two-way interaction between the subgroup of Q5 and vitamin D supplementation (*p* for interaction = 0.14), but it became significant by adjustment with the same seven variables (*p* for interaction = 0.04).

Finally, the effects of vitamin D supplementation on the SHRs of relapse were compared in each quintile of serum PD-L1 levels. No significant effects of vitamin D, compared with placebo, were observed in all quintiles (Q1 HR 1.16, 95% CI 0.30–4.40; Q2 HR 0.39, 95% CI 0.13–1.13; Q3 HR 1.24, 95% CI 0.48–3.22; Q4 HR 1.23, 95% CI 0.46–3.29; and Q5 HR 0.45, 95% CI 0.16–1.29). 

## 4. Discussion

In this clinical study, vitamin D supplementation up-regulated serum PD-L1 levels in the lowest quintile (Q1). This seems to be consistent with the results of experimental research that showed that vitamin D up-regulated expression of PD-L1 in epithelial and myeloid cells [13]. In contrast, vitamin D supplementation down-regulated serum PD-L1 levels in the highest quintile (Q5). Thus, vitamin D may have bimodal functions to increase serum PD-L1 when the serum PD-L1 levels are too low and to decrease serum PD-L1 when the serum PD-L1 levels are too high. However, further research regarding regulation of PD-L1 expression by vitamin D supplementation is needed.

Vitamin D supplementation, compared with placebo, significantly reduced the risk of total death, as well as relapse or death, to one-third in the highest quintile (Q5), but not in other quintiles, i.e., Q1–Q4, and did not change the risk of relapse. Because serum PD-L1 levels increased in an age-dependent manner in the present study and a previous report [19], multivariate adjustment including age was done and showed that they remained significant. In the present study, effects of the interaction between vitamin D and the highest quintile of serum PD-L1 were observed for the outcome of death rather than of relapse. Immune checkpoint inhibitors seem to improve overall survival rather than progression-free survival [20,21,22]. However, how PD-L1 is associated with death rather than relapse of patients has not yet been elucidated. Both the SUNSHINE [23] and AMATERASU [14] trials did not show significance in the primary results, although recent meta-analyses of RCTs suggested that vitamin D supplementation improved the survival of patients with cancer [24,25,26,27]. It has been hypothesized that vitamin D supplementation mainly reduces the risk of total death, at least in part by enhancing anti-cancer immunity and perhaps by keeping cancer tissue dormant by down-regulating serum PD-L1 levels.

This study has several limitations. First, exosomal PD-L1 was not measured in this study. However, not only exosomal PD-L1, but also total plasma PD-L1 was strongly associated with survival of patients with gastric cancer [10]. Second, serum PD-L1 levels were measured only after operation, but not before operation. However, postoperative rather than preoperative levels were reported to be associated with survival of patients with cancer [10]. Third, this study performed an exploratory analysis that was not pre-specified in the original protocol of the AMATERASU trial and must, therefore, be interpreted with caution. Fourth, subgroup analyses of quintiles may increase the probability of type I error due to multiple comparisons. A recent guideline for statistical reporting recommends replacing *p* values with estimates of effects, such as HR and 95% CIs, when neither the protocol nor the statistical analysis plan has specified methods used to adjust for multiplicity [28]. Thus, *p* values were avoided in the present study, except for calculating *p* values for interaction and for changes in serum PD-L1 levels; instead, 95% CIs were used to determine significance. Fifth, because the AMATERASU trial was conducted in Japan, the patients were Asian, most esophageal cancers were squamous cell carcinomas, the incidence of gastric cancer was still relatively high, and the optimal levels of total 25(OH)D and bioavailable 25(OH)D could be different from those in other population groups. Thus, the results of the present study are not necessarily generalizable to other populations. Sixth, the study population included patients with a mixture of cancers with biological and clinical differences.

## 5. Conclusions

Vitamin D supplementation, compared with placebo, may have bimodal functions to increase serum PD-L1 when the serum PD-L1 levels are too low and to decrease serum PD-L1 levels when the serum PD-L1 levels are too high. Vitamin D supplementation, compared with placebo, significantly reduced the risk of all-cause death, as well as relapse or death, to approximately one-third in the highest quintile (Q5), but not in other quintiles, i.e., Q1-Q4. Further studies are needed to explore the mechanisms of bimodal function of vitamin D in the secretion of serum PD-L1 in order to develop potential therapeutic opportunities by supplementation of vitamin D.

## Figures and Tables

**Figure 1 nutrients-13-01987-f001:**
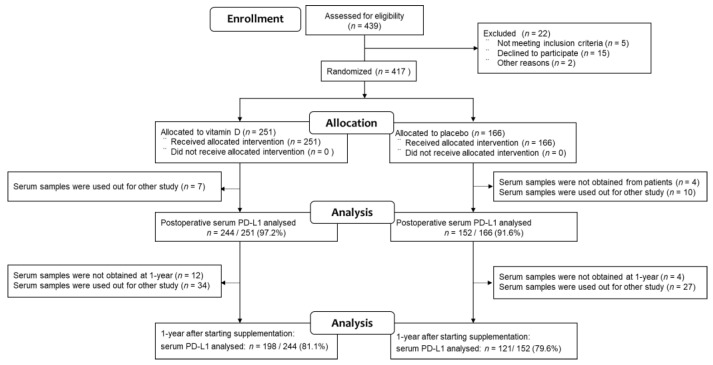
Patient flowchart through the present post hoc analysis.

**Figure 2 nutrients-13-01987-f002:**
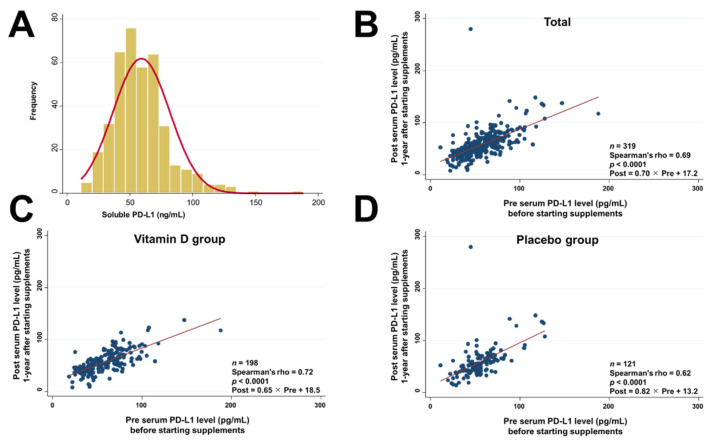
Histogram of serum PD-L1 levels (**A**). Scatter plot between pre serum PD-L1 and post serum PD-L1 levels in all patients (**B**), in the vitamin D group (**C**), and in the placebo group (**D**). Spearman’s rank correlation coefficient (RHO) was used to quantify the strength of the association. The equation was calculated by linear regression analysis.

**Figure 3 nutrients-13-01987-f003:**
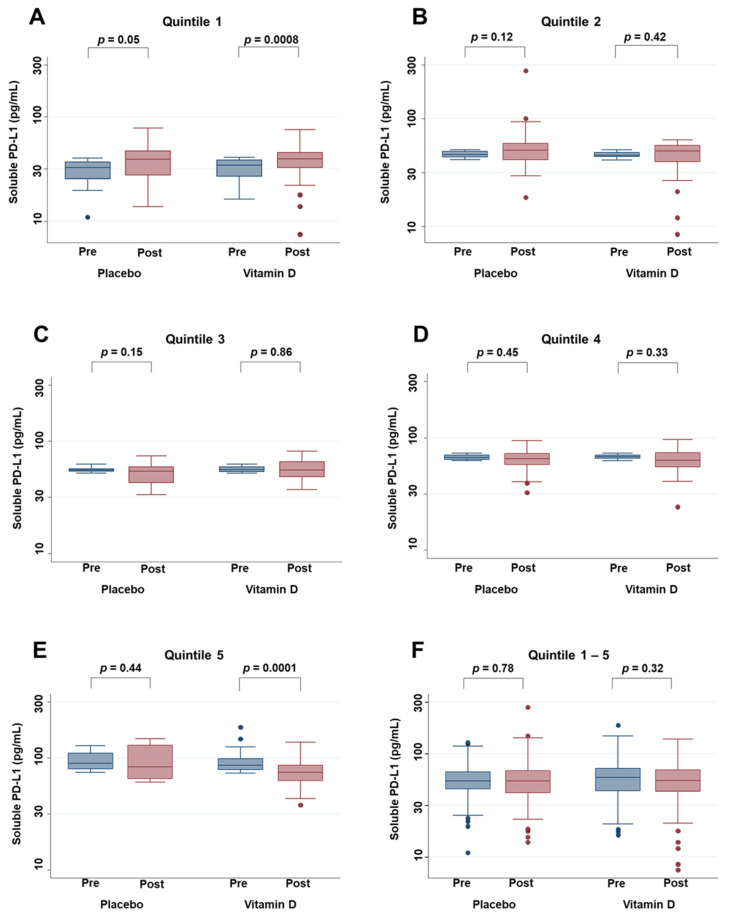
Box plot of changes in serum PD-L1 levels in the placebo group and the vitamin D group compared (**A**) for the subgroup of 1st quintile of PD-L1 levels (Quintile 1), (**B**) for the subgroup of 2nd quintile of PD-L1 levels (Quintile 2), (**C**) for the subgroup of 3rd quintile of PD-L1 levels (Quintile 3), (**D**) for subgroup of 4th quintile of PD-L1 levels (Quintile 4), (**E**) for subgroup of 5th quintile of PD-L1 levels (Quintile 5), (**F**) Sum of 1st–5th quintiles of PD-L1 levels. Pre = after the surgery and just before starting supplements; Post = 1 year after starting supplementation. Changes between pre and post were evaluated with the Wilcoxon signed-rank test.

**Figure 4 nutrients-13-01987-f004:**
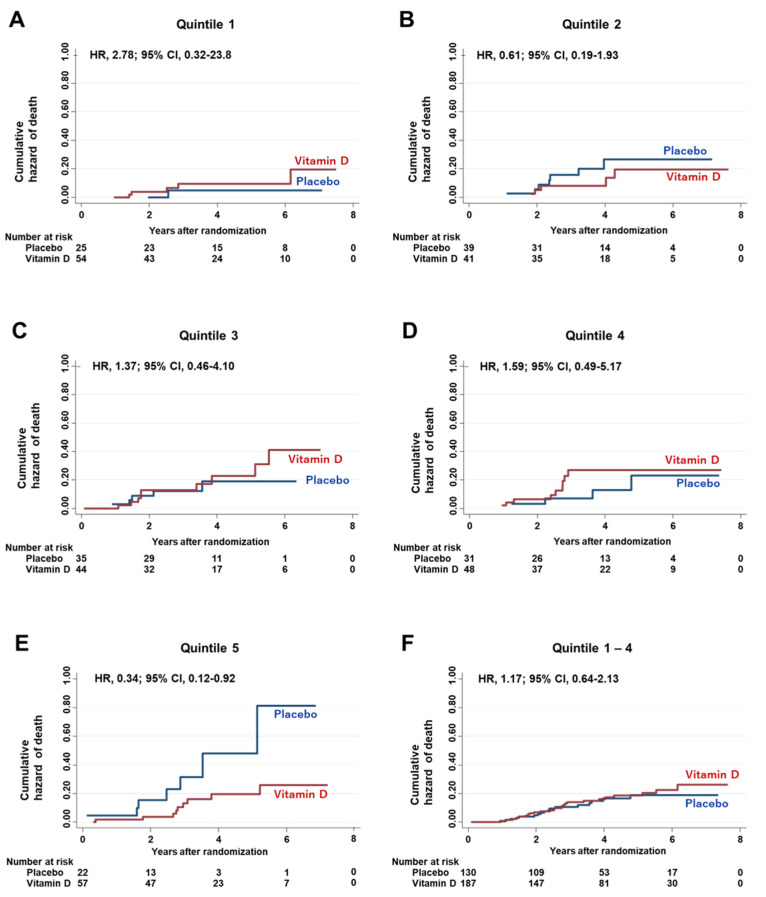
Cumulative hazard curves for death. Nelson–Aalen cumulative hazard curves (**A**) for death in the subgroup of 1st quintile of serum PD-L1 levels (Quintile 1), (**B**) for death in the subgroup of 2nd quintile of serum PD-L1 levels (Quintile 2), (**C**) for death in the subgroup of 3rd quintile of serum PD-L1 levels (Quintile 3), (**D**) for death in the subgroup of 4th quintile of serum PD-L1 levels (Quintile 4), (**E**) for death in the subgroup of 5th quintile of serum PD-L1 levels (Quintile 5), (**F**) for death in the subgroup of the sum of Quintile 1 to Quintile 4 serum PD-L1 levels. HR = Hazard ratio; CI = Confidence interval.

**Figure 5 nutrients-13-01987-f005:**
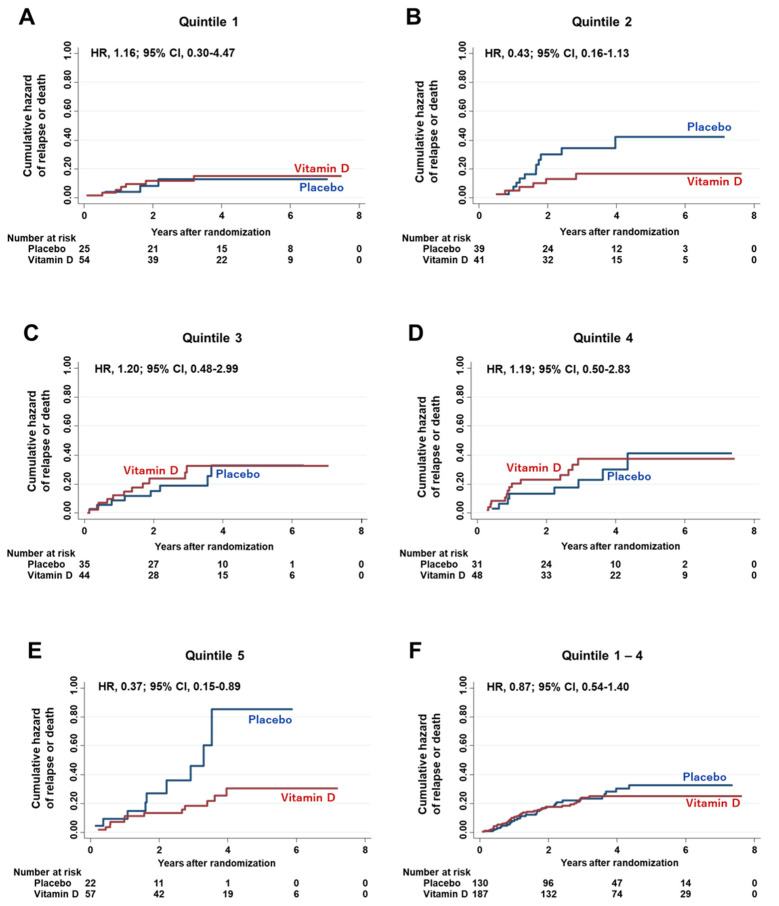
Cumulative hazard curves for relapse or death. Nelson–Aalen cumulative hazard curves (**A**) for relapse or death in the subgroup of 1st quintile of serum PD-L1 levels (Quintile 1), (**B**) for relapse or death in the subgroup of 2nd quintile of serum PD-L1 levels (Quintile 2), (**C**) for relapse or death in the subgroup of 3rd quintile of serum PD-L1 levels (Quintile 3), (**D**) for relapse or death in the subgroup of 4th quintile of serum PD-L1 levels (Quintile 4), (**E**) for relapse or death in the subgroup of 5th quintile of serum PD-L1 levels (Quintile 5), (**F**) for relapse or death in the subgroup of the sum of Quintile 1 to Quintile 4 of serum PD-L1 levels. HR = Hazard ratio; CI = Confidence interval.

**Table 1 nutrients-13-01987-t001:** Patients’ characteristics stratified by Vitamin D vs. Placebo.

*n* = 396	Vitamin D *n* = 244	Placebo *n* = 152
25(OH)D, ng/mL	*n* = 241	*n* = 152
median	21	21
IQR ^a^ (25–75%)	(17–27)	(14.5–26)
25(OH)D, ng/mL 1 year after supplementation	*n* = 208	*n* = 132
median	41	21
IQR ^a^ (25–75%)	(33–55)	(15–27)
Bioavailable 25(OH)D, ng/mL	*n* = 214	*n* = 136
median	1.8	1.6
IQR ^a^ (25–75%)	(1.2–2.8)	(1.1–2.3)
Bioavailable 25(OH)D, ng/mL 1 year after supplementation	*n* = 177	*n* = 117
median	5.0	2.2
IQR ^a^ (25–75%)	(3.4–7.5)	(1.6–3.3)
Sex, *n* (%)	*n* = 244	*n* = 152
Male	171 (70)	94 (62)
Female	73 (30)	58 (38)
Age, y	*n* = 244	*n* = 152
median	67	64
IQR ^a^ (25–75%)	(61–75)	(58–70)
Body mass index (kg/m^2^)	*n* = 242	*n* = 151
median	21.9	22.1
IQR ^a^ (25–75%)	(19.8–24.0)	(20.0–23.7)
History of other cancers, *n* (%)	8 (3)	7 (5)
Comorbid condition, *n* (%)	*n* = 244	*n* = 152
Hypertension	101 (41)	54 (36)
Diabetes Mellitus	44 (18)	21 (14)
Endocrine Disease	32 (13)	16 (11)
Coronary Artery Disease	16 (7)	2 (1)
Stroke	10 (4)	6 (4)
Chronic Kidney Disease	4 (2)	1 (0.7)
Asthma	3 (1)	0 (0)
Orthopaedic disease	1 (0.4)	1 (0.7)
Site of cancer, *n* (%)	*n* = 244	*n* = 152
Oesophagus	22 (9)	15 (10)
Stomach	104 (43)	64 (42)
Small bowel	1 (0.4)	1 (0.7)
Colorectal	117 (48)	72 (47)
Stage, *n* (%)	*n* = 244	*n* = 152
I	113 (46)	61 (40)
II	61 (25)	43 (28)
III	70 (29)	48 (32)
Pathology ^b^		
Adenocarcinoma, *n* (%)	*n* = 244	*n* = 152
Well-differentiated	137 (56)	75 (49)
Moderately differentiated	93 (38)	66 (43)
Poorly differentiated	43 (18)	32 (21)
Signet ring cell	18 (7)	22 (14)
Mucinous	18 (7)	8 (5)
Papillary	11 (5)	4 (3)
Squamous cell carcinoma, *n* (%)	20 (8)	11 (7)
P53 expression, *n* (%)	*n* = 214	*n* = 140
None	35 (16)	28 (20)
Faintly expressed: >0% & <10%	43 (20)	31 (22)
Strongly expressed: ≥10% & <50%	30 (14)	26 (19)
Overexpressed: ≥50%	106 (50)	55 (39)
Vitamin D receptor expression, *n* (%)	*n* = 214	*n* = 140
Quartile 1,	59 (28)	33 (24)
Quartile 2,	54 (25)	35 (25)
Quartile 3,	51 (24)	35 (25)
Quartile 4,	50 (23)	37 (26)
Ki67 expression, *n* (%)	*n* = 214	*n* = 140
Quartile 1,	37 (17)	25 (18)
Quartile 2,	76 (36)	40 (29)
Quartile 3,	30 (14)	29 (21)
Quartile 4,	71 (33)	46 (33)
Adjuvant chemotherapy, *n* (%)	84 (34)	56 (37)

^a^ IQR = Interquartile range. ^b^ Because many patients had multiple histopathologic components, histopathologic subgroups were not mutually exclusive from each other.

**Table 2 nutrients-13-01987-t002:** Patients’ characteristics in subgroups stratified by quintiles of serum PD-L1 levels.

	Total *n* = 396	Q1 *n* = 79	Q2 *n* = 80	Q3 *n* = 79	Q4 *n* = 79	Q5 *n* = 79
Median (IQR ^b^), pg/mL	55.5 (44.2–70.2)	34.4 (26.1–38.7)	45.7 (44.2–49.1)	55.5 (53.4–58.7)	67.3 (64.7–70.2)	86.8 (78.8–103.6)
Intervention						
Vitamin D, *n* (%)	244 (62)	54 (68)	41 (51)	44 (56)	48 (61)	57 (72)
Placebo, *n* (%)	152 (38)	25 (32)	39 (49)	35 (44)	31 (39)	22 (28)
25(OH)D ^c^, ng/mL median (IQR ^b^)						
All	21 (16–27)	22 (17–28)	20 (17–27)	20 (14–25)	22 (17–28)	20 (14–26)
Vitamin D supplementation	21 (17–27)	23 (18–28)	22 (18–28)	19 (15–25)	23 (19–30)	20 (14–26)
Placebo supplementation	21 (15–26)	21 (16–25)	20 (15–27)	22 (13–26)	21 (14–28)	19 (15–26)
25(OH)D ^c^, ng/mL 1 year after supplementation median (IQR ^b^)						
All	33 (21–47)	32 (20–47)	32 (19–42)	32 (21–41)	31 (22–47)	37 (21–55)
Vitamin D supplementation.	41 (33–55)	40 (30–54)	40 (35–54)	40 (33–54)	35 (45–58)	44 (35–60)
Placebo supplementation	21 (15–27)	23 (17–30)	19 (13–28)	22 (16–29)	22 (18–25)	16 (11–23)
Bioavailable 25(OH)D ^c^, ng/mL median (IQR ^b^)						
All	1.71 (1.18–2.59)	1.73 (1.33–2.73)	1.79 (1.28–2.62)	1.75 (1.04–2.31)	1.94 (1.26–2.87)	1.45 (1.00–2.16)
Vitamin D supplementation	1.80 (1.23–2.79)	1.97 (1.17–3.06)	1.95 (1.54–2.74)	1.70 (1.14–2.22)	2.17 (1.57–3.12)	1.52 (1.02–2.48)
Placebo supplementation	1.63 (1.08–2.28)	1.68 (1.53–2.02)	1.71 (1.25–2.42)	1.90 (0.97–2.62)	1.47 (0.97–2.40)	1.26 (0.97–1.65)
Bioavailable 25(OH)D ^c^, ng/mL 1 year after supplementation median (IQR ^b^)						
All	3.62 (2.16–6.20)	3.59 (2.43–5.05)	3.48 (2.00–6.25)	3.35 (1.82–5.71)	3.93 (2.07–6.94)	4.20 (2.16–6.88)
Vitamin D supplementation.	5.05 (3.37–7.51)	4.31 (3.25–7.13)	4.85 (3.60–7.58)	5.21 (3.28–6.96)	5.58 (3.43–8.36)	6.19 (2.82–7.54)
Placebo supplementation	2.25 (1.60–3.32)	2.43 (1.76–3.64)	2.21 (1.27–3.31)	2.22 (1.66–3.74)	2.84 (1.64–3.55)	2.16 (1.41–2.53)
Sex, *n* (%)						
Male	265 (67)	48 (61)	47 (59)	58 (73)	50 (63)	62 (78)
Female	131 (33)	31 (39)	33 (41)	21 (27)	29 (37)	17 (22)
Age, y						
median (IQR ^b^)	66 (60–74)	63 (57–70)	64 (59–73)	64 (57–70)	70 (62–75)	72 (64–78)
Body mass index (kg/m^2^) ^d^						
median (IQR ^b^)	21.9 (19.8–23.8)	22.4 (20.0–24.0)	21.4 (20.0–23.5)	21.7 (20.4–24.2)	21.9 (20.0–23.7)	21.6 (19.2–24.1)
History of other cancers, *n* (%)	15 (3.8)	4 (5.1)	3 (3.8)	5 (6.3)	2 (2.5)	1 (1.3)
Comorbid condition, *n* (%) ^a^						
Hypertension	155 (39)	25 (32)	25 (32)	41 (52)	28 (35)	36 (46)
Diabetes Mellitus	65 (16)	11 (14)	11 (14)	13 (16)	12 (15)	18 (23)
Endocrine Disease	48 (12)	9 (11)	13 (16)	11 (14)	7 (8.9)	8 (10)
Coronary Artery Disease	18 (4.5)	2 (2.5)	3 (3.8)	3 (3.8)	2 (2.5)	8 (10.1)
Stroke	16 (4.0)	1 (1.3)	2 (2.5)	4 (5.1)	3 (3.8)	6 (7.6)
Chronic Kidney Disease	5 (1.3)	0 (0.0)	0 (0.0)	0 (0.0)	2. (2.5)	3 (3.8)
Asthma	3 (0.8)	0 (0.0)	0 (0.0)	0 (0.0)	1 (1.3)	2 (2.5)
Orthopaedic disease	2 (0.5)	0 (0.0)	1 (1.3)	1 (1.3)	0 (0.0)	0 (0.0)
Site of cancer, *n* (%) ^a^						
Oesophagus	37 (9.3)	3 (3.8)	6 (7.5)	7 (8.9)	11 (13.9)	10 (12.7)
Stomach	168 (42.4)	39 (49.4)	37 (46.3)	29 (36.7)	28 (35.4)	35 (44.3)
Small bowel	2 (0.5)	0 (0.0)	0 (0.0)	1 (1.3)	1 (1.3)	0 (0.0)
Colorectal	189 (47.7)	37 (46.8)	37 (46.3)	42 (53.2)	39 (49.4)	34 (43.0)
Stage, *n* (%) ^a^						
I	174 (43.9)	40 (50.6)	29 (36.3)	38 (48.1)	36 (45.6)	31 (39.2)
II	104 (26.3)	19 (24.1)	23 (28.8)	18 (22.8)	21 (26.6)	23 (29.1)
III	118 (29.8)	20 (25.3)	28 (35.0)	23 (29.1)	22 (27.8)	25 (31.6)
Pathology, *n* (%) ^e^						
Adenocarcinoma						
Well-differentiated	212 (53.5)	41 (51.9)	36 (45.0)	47 (59.5)	41 (51.9)	47 (59.5)
Moderately differentiated	159 (40.2)	33 (41 8)	34 (42.5)	25 (31.6)	38 (48.1)	29 (36.7)
Poorly differentiated	75 (18.9)	18 (22.8)	22 (27.5)	11 (13.9)	11 (13.9)	13 (16.5)
Signet ring cell	40 (10.1)	14 (17.7)	9 (11.3)	9 (11.4)	5 (6.3)	3 (3.8)
Mucinous	26 (6.6)	3 (3.8)	7 (8.8)	7 (8.9)	1 (1.3)	8 (10.1)
Papillary	15 (3.8)	3 (3.8)	2 (2.5)	2 (2.5)	4 (5.1)	4 (5.1)
Squamous cell carcinoma, *n* (%)	31 (7.8)	2 (2.5)	5 (6.3)	7 (8.9)	9 (11.4)	8 (10.1)
P53 expression, *n* (%) ^a^						
None	63 (17.8)	13 (19.7)	16 (22.5)	13 (18.3)	13 (17.3)	8 (11.3)
Faintly expressed: >0% & <10%	74 (20.9)	14 (21.2)	19 (26.8)	11 (15.5)	13 (17.3)	17 (23.9)
Strongly expressed: ≥10% & <50%	56 (15.8)	13 (19.7)	14 (19.7)	12 (16.9)	10 (13.3)	7 (9.9)
Overexpressed: ≥50%	161 (45.5)	26 (39.4)	22 (31.0)	35 (49.3)	39 (52.0)	39 (54.9)
Vitamin D receptor expression, *n* (%) ^a^						
Q1,	92 (26.0)	20 (30.3)	22 (31.0)	17 (23.9)	17 (22.7)	16 (22.5)
Q2,	89 (25.1)	15 (22.7)	17 (23.9)	15 (21.1)	19 (25.3)	23 (32.4)
Q3,	86 (24.3)	18 (27.3)	15 (21.1)	20 (28.2)	17 (22.7)	16 (22.5)
Q4,	87 (24.6)	13 (19.7)	17 (23.9)	19 (26.8)	22 (29.3)	16 (22.5)
Ki67 expression, *n* (%) ^a^						
Q1,	62 (17.5)	15 (22.7)	16 (22.5)	10 (14.1)	10 (13.3)	11 (15.5)
Q2,	116 (32.8)	21 (31.8)	24 (33.8)	28 (39.4)	18 (24.0)	25 (32.2)
Q3,	59 (16.7)	15 (22.7)	8 (11.3)	11 (15.5)	15 (20.0)	10 (14.1)
Q4,	117 (33.1)	15 (22.7)	23 (32.4)	22 (31.0)	32 (42.7)	25 (35.2)
Adjuvant chemotherapy, *n* (%) ^a^	140 (35.4)	28 (35.4)	29 (36.3)	28 (35.4)	25 (31.6)	30 (38.0)

^a^ Percentages may not sum to 100% because of rounding. ^b^ IQR = Interquartile range. ^c^ Not measured in some patients. ^d^ Not measured in some patients. Calculated as weight in kilograms divided by height in meters squared. ^e^ Because many patients had multiple histopathologic components, histopathologic subgroups were not mutually exclusive of each other.

## Data Availability

Publicly available datasets were analyzed in this study. This data can be found here: https://upload.umin.ac.jp/cgibin/icdr/ctr_menu_form_reg.cgi?recptno=R000002412 (accessed on 5 April 2019).

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
