# Peer review of "Vitamin D Supplementation Regulates Postoperative Serum Levels of PD-L1 in Patients with Digestive Tract Cancer and Improves Survivals in the Highest Quintile of PD-L1: A Post Hoc Analysis of the AMATERASU Randomized Controlled Trial"

_nutrients, 2021, doi:10.3390/nu13061987_

Round 1

Reviewer 1 Report

An interesting original study about how the use of vitamin D regulates postoperative serum levels of PD-L1 in patients with digestive tract cancer.

The article is very complete, and statistical analysis is very well performed;

Only minor revisions:

Page 2 line 60 you should add: "Vitamin D is a fat-soluble prohormone steroid that has endocrine, paracrine, and autocrine functions. " and cite an article such as: doi: 10.1007/s13668-020-00322-4.

Conclusions should be expanded investigating the further possible developments following the results of this study.

Thank You

Reviewer 2 Report

This was a very well written manuscript with novel findings. The introduction/background was very thorough with a logical flow. I would recommend adding more information at the end of the introduction regarding vitamin D. There is a lack of rationale of why vitamin D was specifically chosen to be explored. Information on the Vitamin's epigenetic potential and immuno modulatory functions would be vital to add, given the study was conducted on cancer patients. 

Line 91: how long after surgery? adding the mean time would be very informative. 

Line 94: who is MM referring to? It may be more appropriate to say it was measured by a member of the research team who was blinded to the groups.

Line 142/145: Remove 'not sampled for some reason'

Line 188: Add pre and post supplementation 

Line 195: Remove whole sample and just inlclude all quintiles 

Reviewer 3 Report

  1. Although the authors have published the blood levels of 25-hydroxyvitamin D it would be extremely helpful and greatly strengthened the manuscript if they had provided this information in this manuscript. They provide baseline levels but there are no data reporting what the blood levels were after receiving 2000 international units vitamin D daily.
  2. Figure 3 can be significantly improved by decreasing the Y-axis so as to expand the data. The few outliers in the figure can be accommodated by making a hashed line on the Y-axis and jumping from 100-200 and 300.

Round 2

Reviewer 3 Report

The authors have responded adequately